# Serum Osteoprotegerin Level Is Negatively Associated with Bone Mineral Density in Patients Undergoing Maintenance Hemodialysis

**DOI:** 10.3390/medicina57080762

**Published:** 2021-07-27

**Authors:** Chia-Wen Lu, Chih-Hsien Wang, Bang-Gee Hsu, Jen-Pi Tsai

**Affiliations:** 1Division of Nephrology, Hualien Tzu Chi Hospital, Buddhist Tzu Chi Medical Foundation, Hualien 97004, Taiwan; noirwen@gmail.com (C.-W.L.); wangch33@gmail.com (C.-H.W.); 2School of Medicine, Tzu Chi University, Hualien 97004, Taiwan; 3Department of Internal Medicine, Division of Nephrology, Dalin Tzu Chi Hospital, Buddhist Tzu Chi Medical Foundation, Chiayi 62247, Taiwan

**Keywords:** bone mineral density, hemodialysis, osteoprotegerin, osteopenia, osteoporosis

## Abstract

Background and Objectives: Osteoprotegerin (OPG), a potent osteoclast activation inhibitor, decreases bone resorption and plays a role in mediating bone mineral density (BMD). Our aim was to evaluate the relationship between BMD and serum OPG in maintenance hemodialysis (MHD) patients. Materials and Methods: Fasting blood samples were obtained from 75 MHD patients. BMD was measured by dual-energy X-ray absorptiometry in lumbar vertebrae (L2–L4). The WHO classification criteria were applied to define osteopenia and osteoporosis. A commercial enzyme-linked immunosorbent assay was used to measure serum OPG values. Results: Among all MHD patients, seven (9.3%) and 20 patients (26.7%) were defined as osteoporosis and osteopenia, respectively. Female patients had lower lumbar BMD than males (*p* = 0.002). Older age (*p* = 0.023), increased serum OPG (*p* < 0.001) urea reduction rate (*p* = 0.021), Kt/V (*p* = 0.027), and decreased body mass index (*p* = 0.006) and triglycerides (*p* = 0.020) were significantly different between the normal, osteopenia, and osteoporosis groups. Lumbar spine BMD was positively correlated with body mass index (BMI) (*p* < 0.001) but negatively correlated with OPG (*p* < 0.001) and age (*p* = 0.003). After grouping patients into T scores < −1 and < −2.5, female sex and OPG (adjusted odds ratio [aOR] 1.022, 95% confidence interval [C.I.] 1.011–1.034, *p* < 0.001) were predictors of T scores < −1, whereas only OPG was predictive of T scores < −2.5 (aOR 1.015, 95% C.I. 1.005–1.026, *p* = 0.004) by multivariate stepwise logistic regression analysis. The areas under the curve for predicting T scores < −1 or < −2.5 were 0.920 (95% C.I. 0.834–0.970, *p* < 0.001) and 0.958 (95% C.I. 0.885–0.991, *p* < 0.001), respectively. Conclusions: Increased serum OPG negatively correlated with lumbar BMD and could be a potential biomarker predictive of osteoporosis in MHD patients.

## 1. Introduction

Chronic kidney disease (CKD) is a health burden affecting nearly 700 million people worldwide that has a higher risk of cardiovascular disease (CVD) [1]. CKD is also a prominent risk factor of fractures related to dysregulated bone metabolism, which is known as CKD-related mineral bone disease, and the risk is higher than the risk in the general population [2,3,4]. Mounting evidence has shown that the prevalence of fractures is higher in CKD patients on dialysis than in pre-dialysis CKD patients [3]. In maintenance hemodialysis (MHD) patients, abnormal low bone mass and density are common, and the examined bone mineral density (BMD) has shown that the prevalence of osteoporosis and osteopenia were 9.5–23% and 16.7–45%, respectively [5,6]. CKD patients with osteoporosis and osteopenia have a substantially increased risk of fractures leading to a huge public health burden worldwide [7].

Multiple risk factors have been reported to be associated with osteoporosis in CKD, including poor nutrition, vitamin-D deficiency, hyperparathyroidism, metabolic acidosis, limited physical activity, and CKD-related metabolic mineral bone disease [8]. Osteoprotegerin (OPG), which belongs to the tumor necrosis factor receptor family, is known as a humoral glycoprotein that binds to the receptor activator nuclear factor κ-B ligand (RANKL) that inhibits osteoclastogenesis [9]. In an in vivo study, mice without OPG exhibited marked osteoporosis as well as decreased bone strength [10]. Moreover, over-expression of OPG resulted in increased bone density in mice as well as inhibition of osteoclast maturation in a dose-dependent manner [11]. In patients with CKD and MHD, serum OPG progressively increased as renal function declined and was positively correlated with inflammatory markers and with survival [12,13,14,15]. Serum OPG levels of dialysis patients were markedly elevated relative to those of controls independent of their serum PTH levels [16]. A comparison of bone histomorphometry in MHD patients showed that there was a negative association between OPG level and trabecular bone volume (BV/TV) [17]. However, the relationship between serum OPG and BMD remained inconclusive. Some studies have shown that there was a positive correlation between BMD and annual percentage changes in BMD and serum OPG levels in MHD patients [18,19]. Another study revealed that there was a negative correlation between serum OPG level and the BMD of the lumbar spine and total hip of female pre-dialysis CKD patients [20] and of the femoral neck of MHD patients [21,22]. Regarding osteoporosis in MHD patients, advanced age and post-menopause status are well-known risk factors for bone loss, but the potential role of OPG in mediating the process of bone loss through its accumulation is not known. Therefore, we conducted this cross-sectional study to determine the role of OPG in the development of osteoporosis and identify the risk factors for bone loss in MHD patients.

## 2. Materials and Methods

### 2.1. Patients

From June 2015 to August 2015 at a single hospital who were >50 years old, were receiving standard weekly 4-h dialysis three sessions using high-flux polysulfone disposable artificial kidneys (FX class dialyzer, Fresenius Medical Care, Bad Homburg, Germany) for ≥3 months were enrolled. Patients were excluded if they were receiving treatments such as bisphosphonates, teriparatide, denosumab or estrogen medications to treat osteoporosis, had a history of lumbar fracture or surgery, acute infection, malignancy, acute cardiovascular disease, or if they declined to provide informed consent. The Research Ethics Committee, Hualien Tzu Chi Hospital, Buddhist Tzu Chi Medical Foundation (IRB106-62-B) approved this study.

### 2.2. Biochemical and Anthropometric Analysis

Before receiving HD therapy, fasting blood samples (approximately 5 mL) were collected and immediately centrifuged at 3000× *g* for 10 min, stored at 4 °C, and analyzed within 1 h after collection. Serum values of biochemical variables were measured by using an autoanalyzer (Siemens Advia 1800, Siemens Healthcare GmbH, Henkestr, Germany). Kt/V and urea reduction ratio (URR) using a formal and single-compartment dialysis urea kinetic model were used to calculate adequacy of HD. Serum OPG (eBioscience Inc., San Diego, CA, USA) and intact parathyroid hormone (iPTH) levels (Abcam, Cambridge, MA, USA) were measured by using a commercially available enzyme immunoassay or enzyme-linked immunosorbent assay, respectively [23]. Body mass index (BMI) was calculated as (body weight)/(body height)^2^ (kg)/(m)^2^ [24].

### 2.3. Bone Mineral Density Measurements

Patients were arranged to examine bone mineral density measurements before HD. Lumbar vertebrate (L2–L4) BMD was measured by using dual-energy X-ray absorptiometry (QDR 4500, Hologic Inc., Marlborough, MA, USA). BMD was expressed as an absolute value (g/cm^2^) and as a T score (deviation from peak BMD) [24]. According to World Health Organization criteria, a lumbar bone T score < −2.5 and −1.0 to −2.5 was used to define osteoporosis and osteopenia, respectively [25].

### 2.4. Statistical Analysis

Continuous variables were expressed as the mean ± standard deviation or as the median and interquartile range according to results of the Klomogorov–Smirnov test. Kruskal–Wallis test or by one-way analysis of variance were used to examine the difference among normal, osteopenia and osteoporosis. Comparisons between male and female patients were analyzed by performing Student’s *t*-test. Categorical variables were expressed as the number of patients and analyzed by the χ^2^ test. Correlation between clinical variables and lumbar BMD were evaluated by linear regression analysis. Variables that were significantly associated with osteopenia or osteoporosis were examined by multivariate stepwise logistic regression analysis. The receiver operating characteristic (ROC) curve was used to calculate the area under the curve (AUC) to identify the optimal cutoff values of OPG for predicting osteopenia or osteoporosis in MHD patients. MedCalc^®^ Statistical Software version 19.7.1 (MedCalc Software Ltd., Ostend, Belgium) was used to analyze. A *p* value < 0.05 was indicative of statistically significant.

## 3. Results

In this study, there were 48 (64%), 20 (26.7%), and 7 (9.3%) in the normal, osteopenia, and osteoporosis groups, respectively (Table 1). Compared with the normal group, patients in the osteopenia or osteoporosis groups were older (63.94 ± 8.89 vs. 70.25 ± 8.84 vs. 68.43 ± 7.09; *p* = 0.023) and had more females (35.4% vs. 75% vs. 85.7%, *p* = 0.002), higher levels of OPG (231.27 ± 82.47 vs. 413.87 ± 118.05 vs. 665.37 ± 191.45, *p* < 0.001), higher URR (0.73 ± 0.04 vs. 0.75 ± 0.04 vs. 0.76 ± 0.04; *p* = 0.021), and higher Kt/V (1.30 ± 0.16 vs. 1.41 ± 0.17 vs. 1.44 ± 0.17; *p* = 0.027) but lower BMI (25.92 ± 5.01 vs. 23.61 ± 3.94 vs. 20.26 ± 2.26, *p* = 0.006). Values of lumbar BMD and T scores were significantly lower for the female MHD patients than for the males (Figure 1). Serum levels of alkaline phosphatase or iPTH showed no significant differences among these three groups.

Linear correlation analysis showed that BMI (*r* = 0.40, *p* < 0.001; *r* = 0.41, *p* < 0.001) was positively correlated, whereas age (*r* = −0.33, *p* = 0.004; *r* = −0.34, *p* = 0.003) and OPG (*r* = −0.60, *p* < 0.001, *r* = −0.58, *p* < 0.001) was negatively correlated with lumbar T score and BMD, respectively (Figure 2).

Patients were then grouped by T score between −1 and −2.5 (osteopenia and osteoporosis, Table 2) and −2.5 (osteoporosis, Table 3). After adjustment for factors significantly associated with T scores < −1 and −2.5 in the univariate logistic regression analysis, the serum OPG level (adjusted odds ratio (aOR) 1.022, 95% confidence interval (C.I. 1.010−1.045, *p* = 0.002)) and female sex (aOR 13.37; 95% C.I., 2.049−87.30; *p* = 0.007) were identified as independent predictors of T scores less than −1 (Table 2), whereas serum OPG (aOR 1.015; 95% C.I., 1.005−1.026; *p* = 0.004) was the single significant predictor for T scores < −2.5 by multivariate stepwise logistic regression analysis (Table 3).

The ROC curve analysis showed that the best serum cutoff values of OPG to predict osteopenia + osteoporosis and osteoporosis alone of MHD patients were 388.38 pg/mL and 394.73 pg/mL, with AUCs of 0.920 (95% C.I., 0.834−0.970; *p* < 0.001) and 0.958 (95% C.I., 0.885−0.991; *p* < 0.001), respectively (Figure 3).

## 4. Discussion

In this study, we found that female MHD patients had lower BMDs than males and advanced age was positively correlated and serum OPG and BMI were negatively associated with lumbar BMD in MHD patients. Moreover, OPG could be a potential biomarker for the diagnosis of osteoporosis with or without osteopenia of MHD patients.

Among CKD patients, evidence had shown an abnormal quantity and quality of bone metabolism with a high risk of fracture and poor long-term survival [5,6,7,26,27]. Therefore, the 2017 Kidney Disease: Improving Global Outcomes (KDIGO) guideline recommended dual-exergy X-ray for assessing and managing the risk of fracture in CKD patients [26]. There are multiple known risk factors related to lower BMD in CKD, especially in dialysis patients. In these studies, female and old age were consistently found to be risk factors related to osteoporosis of advanced CKD and dialysis patients [28,29,30]. Osteo-sarcopenia was a newly accepted concept of a link between osteoporosis and sarcopenia that together could lead to a higher risk of frailty [28,31]. In a longitudinal study, lower BMD was correlated with low BMI and poor renal outcomes in non-dialysis CKD patients [28]. Similarly, BMI and especially the skeletal muscle mass index were found to significantly affect the lumbar spine BMD of MHD patients [31]. Consistent with these studies, we found that MHD had a higher percentage of osteopenia (26.7%) and osteoporosis (9.3%) and that lumbar spine BMD was negatively associated with age and female sex but positively correlated with BMI.

OPG, produced from the CV system and bone, was initially known as osteoclastogenesis inhibitory factor and is a member of the tumor necrosis factor receptor family. OPG has a role in regulating the process of osteoclastogenesis and vascular calcification [9,32,33]. Mice with OPG knockout exhibited increased numbers of osteoclast cells and marked femoral bone loss along with destruction of growth plate, lack of trabecular bone, and abnormal cortical bones as well as dramatically decreased BMD, bone mineral content, stiffness, and strength relative to heterozygous and wild-type controls [10]. Contrarily, transgenic mice overexpressing OPG exhibited markedly increased bone density of long bones, vertebrae, and pelvis associated with decreased osteoclast-mediated bone resorption [11]. Moreover, recombinant OPG inhibited osteoclastogenesis in vitro, increased bone density in vivo dose dependently, and protected rats against ovariectomy-associated bone loss [11]. In addition to exhibiting significantly decreased trabecular and cortical bone density within long bones and vertebrae, OPG-deficient mice showed increased medial arterial calcification of the aorta and renal arteries with endogenous OPG/OPG ligand/RANK expression within the smooth-muscle layer on the other hand [32,33]. A meta-analysis showed that CKD patients (including those on dialysis) had a 1.04 higher risk of CV mortality associated with an increase of 1 pmol/L in OPG concentration, which indicated a positive relationship between OPG levels and vascular calcification through mediating the process of osteoclastogenesis, osteoblast activation and osteoclast-like formation [34]. Increased serum OPG level has been associated with a 1.13 higher risk of self-report and/or prevalent vertebral fracture independent of CKD stages and sex, which supposed a compensatory role of OPG with increasing bone loss of CKD [35]. From these evidence, it seems that increased OPG production and release with increased serum levels of OPG could mediate the process of bone loss but have a role in pathological arterial calcification that results in poor long-term prognosis [12,15].

Osteoporosis is known as a metabolic bone disease characterized by unequilibrated bone formation and resorption. From a cross-sectional study of about 1000 healthy participants, OPG levels correlated negatively with tartrate-resistant acid phosphatase-5b (a bone resorption marker) but positively with osteocalcin (a bone formation marker), which indicated that OPG tended toward bone formation more than bone resorption [36]. However, there have been conflicting evidence demonstrating serum OPG levels with regards to BMD in CKD patients. Our findings supported these results such as studies have shown a negative correlation between the serum OPG level and BMD of the lumbar spine and total hip of female pre-dialysis CKD patients [20] and of the femoral neck BMD of MHD patients [21,22]. Furthermore, by observing bone histomorphometry in MHD patients, Barreto et al. demonstrated that independent determinants of osteoporosis determined by trabecular bone volume (BV/TV) were the sRANKL/OPG ratio and length of amenorrhea, which indicated bone loss through the OPG/OPG ligand/RANK signal pathway [17]. However, in contrast, Nakashima et al. and Moldovan et al. reported a positive correlation between BMD and serum OPG levels in MHD patients [18,19]; moreover they showed that the annual percentage change in BMD correlated positively with OPG level [18]. Accordingly, we postulated that OPG might increase with an increase in bone turnover, perhaps as a compensation to prevent bone loss, but the definite mechanism needed further studies. Together with these studies, we found that OPG was independently and negatively associated with lumbar BMD in MHD patients after adjusting the covariates with optimal cutoff values of 388.38 pg/mL and 394.73 pg/mL associated with T scores < −1 (osteopenia and osteoporosis) or −2.5 (osteoporosis), respectively.

There were limitations of this study. First, it was a cross-sectional design. Secondly, the number of MHD patients was limited. Thirdly, we did not measure serum markers of bone formation or resorption. Therefore, to predict bone loss of MHD patients with serum OPG level needed further longitudinal studies to establish a cause and effect.

## 5. Conclusions

Our study showed that along with old age and female sex, there was a negative relationship between lumbar BMD and serum OPG of MHD patients. These findings indicated that OPG could be a biomarker and a modulator in the regulation of bone loss. These potential characteristics require confirmation in specifically designed studies in MHD patients.

## Figures and Tables

**Figure 1 medicina-57-00762-f001:**
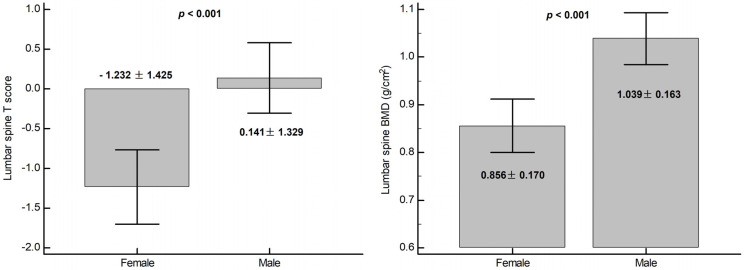
Different values of lumbar spine BMD and T score between female and male.

**Figure 2 medicina-57-00762-f002:**
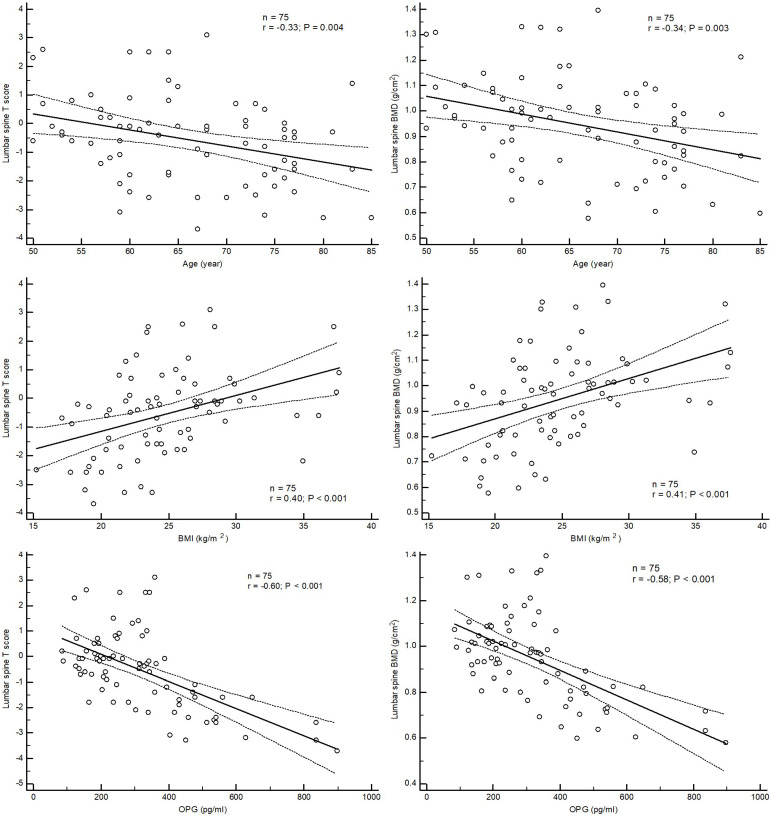
Correlation between T score and BMD of lumbar spine and age, BMI and OPG.

**Figure 3 medicina-57-00762-f003:**
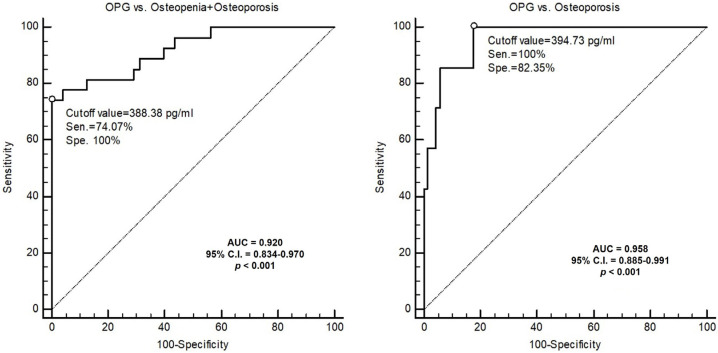
ROC curve of OPG to predict osteopenia (T score < −1) and osteoporosis (T-score < −2.5) or osteoporosis.

**Table 1 medicina-57-00762-t001:** Clinical characteristics according to different lumbar T-score cut-off points (normal, osteopenia, and osteoporosis) of the 75 hemodialysis patients.

Characteristics	All Patients (*n* = 75)	Normal (*n* = 48)	Osteopenia (*n* = 20)	Osteoporosis (*n* = 7)	*p* Value
Age (years)	66.04 ± 9.08	63.94 ± 8.89	70.25 ± 8.84	68.43 ± 7.09	0.023 *
Female, *n* (%)	38 (50.7)	17 (35.4)	15 (75.0)	6 (85.7)	0.002 *
Diabetes mellitus, *n* (%)	37 (49.3)	27 (56.3)	8 (40.0)	2 (28.6)	0.244
Hypertension, *n* (%)	33 (44.0)	21 (43.8)	9 (45.0)	3 (42.9)	0.993
Hemodialysis duration (months)	49.00 (21.00–110.00)	52.00 (19.00–114.00)	46.00 (24.00–108.00)	98.00 (17.00–255.00)	0.658
Body mass index (kg/m^2^)	24.78 ± 4.86	25.92 ± 5.01	23.61 ± 3.94	20.26 ± 2.26	0.006 *
Lumbar bone mineral density (g/cm^2^)	0.95 ± 0.19	1.06 ± 0.13	0.79 ± 0.07	0.65 ± 0.05	<0.001 *
Lumbar T-score	−0.55 ± 1.53	0.34 ± 1.07	−1.83 ± 0.56	−3.01 ± 0.43	<0.001 *
Systolic blood pressure (mmHg)	139.89 ± 26.26	142.46 ± 24.68	141.00 ± 28.39	119.14 ± 24.90	0.086
Diastolic blood pressure (mmHg)	73.64 ± 14.00	74.77 ± 14.70	74.65 ± 12.20	63.00 ± 10.38	0.107
Albumin (mg/dL)	4.10 (3.80–4.40)	4.10 (3.90–4.40)	4.10 (3.85–4.25)	3.70 (3.60–5.20)	0.728
Total cholesterol (mg/dL)	145.31 ± 33.99	142.75 ± 34.85	149.95 ± 31.97	149.57 ± 36.82	0.691
Triglyceride (mg/dL)	126.00 (90.00–200.00)	157.50 (104.50–219.50)	97.00 (84.00–144.00)	101.00 (53.00–131.00)	0.020 *
Glucose (mg/dL)	135.00 (110.00–185.00)	143.50 (110.00–196.00)	124.00 (107.00–159.25)	132.00 (103.00–142.00)	0.360
Blood urea nitrogen (mg/dL)	57.55 ± 13.12	55.92 ± 12.48	59.30 ± 14.35	63.71 ± 13.15	0.269
Creatinine (mg/dL)	9.11 ± 1.83	9.33 ± 1.85	8.75 ± 1.82	8.64 ± 1.77	0.387
Alkaline phosphatase (U/L)	88.80 ± 36.06	82.25 ± 33.87	96.90 ± 37.15	110.57 ± 39.56	0.077
Total calcium (mg/dL)	8.96 ± 0.73	8.90 ± 0.67	9.19 ± 0.81	8.75 ± 0.85	0.231
Phosphorus (mg/dL)	4.49 ± 1.23	4.59 ± 1.10	4.42 ± 1.44	4.06 ± 1.50	0.543
Intact parathyroid hormone (pg/mL)	231.68 ± 185.34	198.68 ± 174.60	287.26 ± 192.13	299.21 ± 208.87	0.119
Osteoprotegerin (pg/mL)	320.48 ± 172.17	231.27 ± 82.47	413.87 ± 118.05	665.37 ± 191.45	<0.001 *
Urea reduction rate	0.74 ± 0.04	0.73 ± 0.04	0.75 ± 0.04	0.76 ± 0.04	0.021 *
Kt/V (Gotch)	1.34 ± 0.17	1.30 ± 0.16	1.41 ± 0.17	1.44 ± 0.17	0.027 *

Values for continuous variables given as means ± standard deviation and test by one-way analysis of variance; variables not normally distributed given as medians and interquartile range and test by Kruskal–Wallis analysis. * *p* < 0.05 was considered statistically significant after Kruskal–Wallis analysis or one-way analysis of variance. Kt/V, fractional clearance index for urea.

**Table 2 medicina-57-00762-t002:** Risk factor for diagnosis of osteopenia and osteoporosis (T score < −1).

Variable	OR	95% CI	*p* Value	aOR	95% CI	*p* Value
Age (years)	1.080	1.019–1.144	0.001 *	—	—	—
Female	6.382	2.161–18.85	<0.001 *	13.37	2.049–87.30	0.007 *
Diabetes mellitus	0.458	0.174–1.204	0.113	—	—	—
Hypertension	1.029	0.398–2.658	0.954	—	—	—
Hemodialysis duration (months)	1.004	0.997–1.012	0.237	—	—	—
Body mass index (kg/m^2^)	0.844	0.792–0.960	0.010 *	—	—	—
Albumin (mg/dL)	0.727	0.215–2.462	0.609	—	—	—
Total cholesterol (mg/dL)	1.006	0.992–1.020	0.385	—	—	—
Triglyceride (mg/dL)	0.993	0.987–1.0	0.041 *	—	—	—
Glucose (mg/dL)	0.995	0.987–1.003	0.203	—	—	—
Blood urea nitrogen (mg/dL)	1.028	0.990–1.068	0.155	—	—	—
Creatinine (mg/dL)	0.825	0.626–1.086	0.170	—	—	—
Kt/V (Gotch)	23.00	2.021–261.7	0.012 *	—	—	—
Alkaline phosphatase (U/L)	1.014	1.001–1.028	0.042 *	—	—	—
Total calcium (mg/dL)	1.414	0.729–2.741	0.306	—	—	—
Phosphorus (mg/dL)	0.835	0.561–1.241	0.372	—	—	—
Intact parathyroid hormone (pg/mL)	1.003	1.000–1.005	0.043	—	—	—
Osteoprotegerin (pg/mL)	1.020	1.010–1.029	<0.001 *	1.022	1.011–1.034	<0.001 *

Multivariate stepwise logistic regression adjusted by age, gender, diabetes mellitus, body mass index, triglyceride, alkaline phosphatase, intact parathyroid hormone, Kt/V, and osteoprotegerin. * *p* < 0.05 was considered statistically significant. OR, odds ratio; CI, confidence interval; aOR, adjusted odds ratio; Kt/V, fractional clearance index for urea.

**Table 3 medicina-57-00762-t003:** Risk factor for diagnosis of osteoporosis (T score < −2.5).

Variable	OR	95% CI	*p* Value	aOR	95% CI	*p* Value
Age (years)	1.033	0.947–1.127	0.465	—	—	—
Female	6.750	0.771–59.11	0.085	—	—	—
Diabetes mellitus	0.377	0.068–2.080	0.263	—	—	—
Hypertension	0.950	0.197–4.574	0.949	—	—	—
Hemodialysis duration (months)	1.010	1.000–1.020	0.056	—	—	—
Body mass index (kg/m^2^)	0.691	0.518–0.993	0.012 *	—	—	—
Albumin (mg/dL)	1.716	0.262–11.24	0.573	—	—	—
Total cholesterol (mg/dL)	1.004	0.982–1.027	0.726	—	—	—
Triglyceride (mg/dL)	0.993	0.981–1.005	0.251	—	—	—
Glucose (mg/dL)	0.990	0.972–1.008	0.269	—	—	—
Blood urea nitrogen (mg/dL)	1.042	0.979–1.109	0.194	—	—	—
Creatinine (mg/dL)	0.849	0.539–1.337	0.479	—	—	—
Kt/V (Gotch)	25.40	0.649–994.8	0.084	—	—	—
Alkaline phosphatase (U/L)	1.016	0.997–1.035	0.106	—	—	—
Total calcium (mg/dL)	0.636	0.212–1.909	0.419	—	—	—
Phosphorus (mg/dL)	0.707	0.355–1.407	0.323	—	—	—
Intact parathyroid hormone (pg/mL)	1.002	0.998–1.006	0.316	—	—	—
Osteoprotegerin (pg/mL)	1.015	1.005–1.026	0.004 *	1.015	1.005–1.026	0.004 *

Multivariate stepwise logistic regression adjusted by age, gender, diabetes mellitus, body mass index, hemodialysis duration, Kt/V, and osteoprotegerin. * *p* < 0.05 was considered statistically significant. OR, odds ratio; CI, confidence interval; aOR, adjusted odds ratio; Kt/V, fractional clearance index for urea.

## Data Availability

The data presented in this study are available on request from the corresponding author.

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
