# Peer review of "Serum Osteoprotegerin Level Is Negatively Associated with Bone Mineral Density in Patients Undergoing Maintenance Hemodialysis"

_medicina, 2021, doi:10.3390/medicina57080762_

Round 1
Reviewer 1 Report
I find this study of interest since there is stil a debate regarding the effects of OPG levels elevation on in MHD patients.
The article is well written, the methods are appropriate for the aim of the study, the results are clearly explained and the findings properly described in the context of the published literature.
I have a concern, although:
the first sentence in the Abstract stipulates that " OPG... decreases bone resorption and positively affects bone mineral density (BMD)". It is presented like a fact. Later in the article, the debate regarding the actual role of OPG in MHP is rised: "However,the relationship between serum OPG and BMD remained inconclusive". In the conclusions, the result of the study is that ", there was a negative relationship between lumbar BMD and serum OPG of MHD patients".
I consider that the first sentence in the abstract should be changed.
Good work, however!
Author Response
I find this study of interest since there is stil a debate regarding the effects of OPG levels elevation on in MHD patients.
The article is well written, the methods are appropriate for the aim of the study, the results are clearly explained and the findings properly described in the context of the published literature.
I have a concern, although:
the first sentence in the Abstract stipulates that " OPG... decreases bone resorption and positively affects bone mineral density (BMD)". It is presented like a fact. Later in the article, the debate regarding the actual role of OPG in MHP is rised: "However,the relationship between serum OPG and BMD remained inconclusive". In the conclusions, the result of the study is that ", there was a negative relationship between lumbar BMD and serum OPG of MHD patients".
I consider that the first sentence in the abstract should be changed.
Ans: Thanks for your recommendations. We would revise the abstract according to your suggestions.

Reviewer 2 Report
This article tackles an important issue in HD patiens- the CKD-MBD. While the results are interesting, they are some points that need to be clarified:
- in the methods section, the authors note that " Patients were excluded if they were receiving treatments of osteoporosis (bisphosphonates,
teriparatide, or estrogen medications..." - the authors should check if denosumab was used since it interferes with RANKL - the authors should explain more in detail the discrepancy between the protective role of osteoprotegerin in experimental models and the deleterious effect in HD patients as was depicted in their study
"
Author Response
Comments and Suggestions for Authors
This article tackles an important issue in HD patiens- the CKD-MBD. While the results are interesting, they are some points that need to be clarified:
- in the methods section, the authors note that " Patients were excluded if they were receiving treatments of osteoporosis (bisphosphonates,
teriparatide, or estrogen medications..." - the authors should check if denosumab was used since it interferes with RANKL
Ans: Thanks for your suggestions. We re-examined the medical records of patients and confirmed that participants did not receive denosumab. We will include this description in this article.
- the authors should explain more in detail the discrepancy between the protective role of osteoprotegerin in experimental models and the deleterious effect in HD patients as was depicted in their study
Ans: Thanks for your suggestions. We revised this article and explained clearer about the discrepancy between OPG levels and its harmful effects in HD patients in the Discussion section.

Round 2
Reviewer 2 Report
The authors responded to all the questions.
The last phrase from the discussion section is unclear. Please re prahse